# Urbanization and flood risk analysis using geospatial techniques

Raphael Ane Atanga[1]*, Vitus Tankpa[2], Isaiah Acquah[1]

1 Department of Geography Education, Faculty of Social Sciences Education, University of Education, Winneba, Ghana, 2 Environmental Impact Division, Ghana Energy Commission, Department of Environmental and Natural Resources, Faculty of Development Studies, Presbyterian University College, Abetefi, Ghana

* raatanga@uew.edu.gh

## Abstract

This research investigates the relationship between urbanization as a land use/land cover change and the increased flood disasters in Accra. Understanding this relationship will provide evidence for urban development planners, policy makers and flood managers to coordinate in responding to the problems effectively. This study maps and analyzes the changes in urbanization from 1991 to 2015. The research reviews the trends of flood events in Greater Accra and analyzes the relationship between the pattern of urbanization and the increase in flood disaster events from 1991 to 2015. The research revealed that there was an increase in urban land use/land cover change of up to 95.51% and 129.14% in the periods 1991–2002 and 2002–2015 respectively. The pattern of urbanization took place in an unplanned style, where physical developments in waterways became high. The findings show that the pattern of flood disasters increased from 1991 to 2015 with evidence showing two years having repeated flood events.

## 1 Introduction

This study analyses the relationship between urbanization as a land use/land cover change and flood disasters in Accra using geospatial techniques. Land use/land cover changes are major concerns for environment and development planning, especially when the changes have negative impacts on society [1]. Land use/land cover changes can come from natural and human induced processes which can destroy and replace existing land use/land cover surfaces. Existing land use/landcover surfaces as ecosystems, are difficult and expensive to remediate when destroyed. Land use/land cover changes can be influenced by political, economic, demographic and environmental factors [2]. Changes in political decisions and development policies may favour and discourage people from living in urban areas. For instance, where economic and physical development focused more on the provision of infrastructure, services and quality of life in urban areas than rural development, people may be attracted to migrate to urban settings [3]. The migrants would need more houses and urban land spaces which could eat up other land cover surfaces. Economic reasons for changes in land use/land cover follow similar trend of argument where economic activities were predominantly urban, thereby attracting the concentration of people in urban environs. In the 1990s, for example,

**Competing interests:** The authors have declared that no competing interests exist.

countries that experienced trade liberalization and structural adjustments policies focused more on urban development which subsequently attracted the working population to cities and towns leading to urbanization [1]. Land use/land cover changes including land degradation, natural disasters and insecurity may drive people to urban areas leading to increase in demand for urban dwellings and environmental changes. Demographic changes as in population increase in rural and urban settings can lead to a need for more land for housing and development which may trigger changes in land cover and land use to urban land surface. Of all developmental activities that bring about land use land cover changes, urbanization is one of the prominent changes that the world has been experiencing over the years that lead to spiral social and environmental challenges [4]. Urbanization is the uncontrolled expansion of urban land uses emanating from development. Urbanization is particularly a problem of developing countries, especially in Africa where the percentage of urban population is less than 50, which means that the continent is rapidly urbanizing [5]. Changes in society that accelerate urbanization may include demographic, economic and technological advancements, leading to the quest to dwell in urban environments.

Impacts of urbanization on environment in developing nations includes modernization and improvement in quality of life but it can also result in environmental problems including degradation of ecosystems, pollution and poor sanitation, solid waste, slum development and increase in flood disaster risk [5–8]. Urbanization and its impacts in Ghana are a serious concern for planners, development practitioners and policy makers alike as major cities including Accra and Kumasi keep expanding uncontrollably over the years. About 50% of the Ghana population is urban dwellers and most recent research shows that this figure will reach 70% by the year 2050 [4]. Factors that explain the reasons for the increase of urban dwellers in Ghana are related to those mentioned modernization, rural urban migration, globalization and environmental factors [9]. Greater Accra is the capital of Ghana with a current population of over 4 million of which over 90% is urban and the total population is expected to exceed 10 million in the next 20 years [4]. The impacts of urbanization have consequences on land use land cover changes, environmental degradation, waste management challenges, conversion water bodies into urban developments and encroachment on waterways resulting increase in flood disaster risk [10]. This is particularly for Accra where urban sprawl is a problem and a major cause of environmental degradation, insanitary conditions, pollution, and annual flood disasters [11]. The rate of urbanization and its related challenges of development keep increasing and require further solutions. Flood disaster risk has been a chronic annual problem in Accra since the 1990s with serious impacts on lives, property and environment [7, 12–14]. The spate of flood problems seems to coincide with the period in which urbanization in Accra started gaining momentum [3, 15]. However, relationship between flood disaster risk and urbanization in Accra is not yet well examined using remote sensing and geographic information systems (GIS).

The development of remote sensing and geographic information systems technologies has brought possibilities to study the spatio-temporal changes of land use land cover that allow researchers to understand in quantitative and qualitative terms, the extent of changes in urbanization and its relationship with other urban environmental problems [15, 16]. Particularly, GIS plays a crucial role in mapping, modeling and spatial analysis of land use land cover changes which includes urbanization and flood disasters [17–19]. [20] reviewed successful application of remote sensing and GIS of all stages in flood disaster risk management (pre-flood, during flood and post-flood event management phases). Several studies about urbanization and flood disaster risk have been conducted in Accra using GIS and remote sensing but none of them has focused on mapping and explaining the relationship between urbanization and flood disasters. [3] indicates that urbanization of Accra makes the city vulnerable to

climate change impacts. [7] studied urban planning problems and flood disasters in Accra and concludes on poor urban planning as one reason for flood disasters in the city.

For instance, using GIS [21], modeled flood risk zones in Accra and contended that over 40% is it between high and medium flood risk zones. Other scholars employed GIS and remote sensing techniques to investigate vulnerability of slums in Accra to flooding and environmental degradation using social status, demographics and health as yardsticks [22]. [23] further applied GIS to model and predict the future extent of flood risk in Accra. [24] applied a triangulated irregular network and GIS to analyse reservoirs for flood management in Accra.

[9] examine the spatio-temporal land use land cover change patterns that occurred in Accra between 1985 and 2010 using remote sensing and GIS techniques. [25] suggest that risk of flooding to settlements in Accra will keep increasing the next years. The most recent study by [4] suggests that urbanization keeps increasing at the expense of other land use land cover surfaces and this could reach about 70% in the year 2025.

The existing urban planning policies and flood risk management measures in Accra have been unsuccessful in stopping urbanization and floods in Greater Accra [26]. The Town and Country Planning department exists with a history of land use planning policies and measures for responding to urbanization and floods but have not been successful yet [7]. The main flood management interventions in Accra include structural and non-structural measures. The structural measures are engineering approaches, mainly channelization which focuses construction and maintenance of drains to convey storm waters to sea. These are physical structures with an objective prevent and control flood hazards and vulnerabilities. The Korle Lagoon Restoration Project (KLERP) and the Accra Sanitary and Storm Drainage Alleviation Project 2013 (the Conti Project) are examples of structural flood management interventions.

The non-structural measures are policies and programmes with an objective to save lives and property through flood preparedness, emergency response, recovery and mitigation measures. Institutions such as NADMO, Ghana Meteorological Agency, security agencies and non-state organizations play various roles in flood management of the study site. The flood preparedness measures include public education on safety measures as well as eviction and relocation of settlements on waterways [27]. During flood events, early warning of floods for safety and evacuation further support to minimize flood impacts. There are also relief interventions and clean-up exercises after flood events to minimize post-flood impacts. These measures together with flood mitigation interventions exist but the menace of floods keeps increasing.

Seemingly, flooding is a big problem in Accra and there is a knowledge gap about the relationship between changes in urbanization and flood disasters from Remote sensing and GIS analysis. Remote sensing and GIS can show temporal and spatial changes in urbanization to enhance quantitative and qualitative understanding of the changes and the relationship with flood disaster risks in Accra. This study seeks to map and analyze the changes in urbanization for the periods of 1991 to 2015 (1991–2002 and 2002–2015) and the relationship of these changes with increase in flood disasters in Accra within the period. Establishing this relationship would help to map and understand the relationship between urbanization and flood disaster risk in the city to inform urban planning and development in the future.

This research has three specific research objectives. The first objective is to map and analyze the trends of urbanization as a land use land type in Greater Accra for the periods of 1991–2002 and 2002–2015 using geospatial techniques. The second objective is to map flood risk zones, identify the flood disaster events between 1990 and 2015 and analyze the trends of flood disaster risk in Greater Accra. The third objective is to analyze the relationship between the pattern of urbanization and the increase in flood disaster events from 1990 to 2015. The aim of

this research is to contribute to research, policy and practical management of urbanization and flood disaster risk in the study site and other areas with similar conditions.

## 2 Materials and methods

### 2.1 Description of the study area

The Greater Accra Region of Ghana was chosen for the research due to its increasing urbanization and chronic annual flood disasters in recent years. [3] provided extensive mapping and analysis of uncontrolled urbanization in Accra. Accra in the Metropolitan Assembly (AMA) is the capital of the Greater Accra Region and the capital city of Ghana. It is the centre of development and the headquarters of international organizations that link Ghana to rest of the world.

Accra is between Latitude 5º 40'N and between Longitude 00º 34'W and Longitude 00º 29'E (Fig 1). The region has about 780 cubic millimeters of average annual rainfall with temperature fairly stable from 24.7˚C in August to 28˚C in March with an annual average of 26.8˚C. Located along the Gulf Guinea Coast, the northern part of Accra is bordered by the Akwampim ranges with Tema Municipal Assembly bordering eastern part. The topography of Accra is undulating with generally lowland dissected by 7 drainage basins. Namely, Chemu, Osu, Songo Mokwe, Lafa, Sakumo, Densu, Kpeshie and Korle basins. The Korle basin experiences high runoff resulting in frequent flood disasters. The region lies within -40 meters below and 400 meters above sea level. The surroundings of the drainage basins in low lying areas experience floods. The flood prone communities are located in very high to medium flood risk zones. Studies have established that these communities regularly had flood events leading to disasters [3, 21].

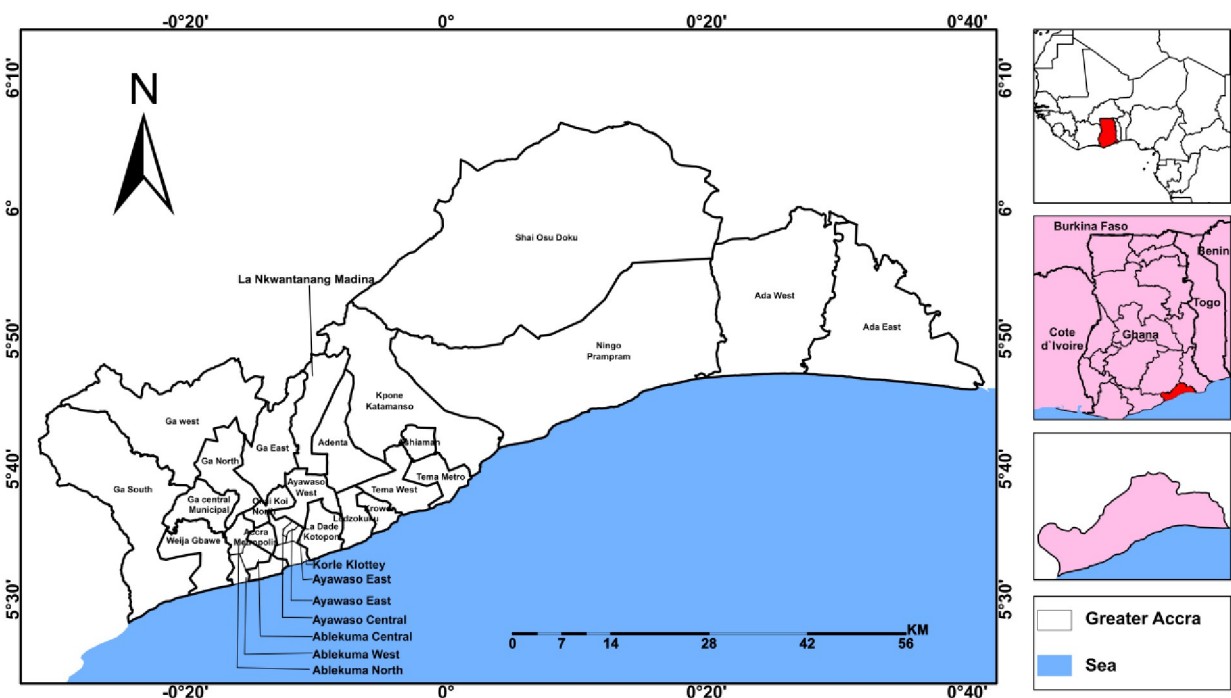

**Fig 1. Location of study area. Source**: Fieldwork, 2020.

**Table 1. Characteristics of the satellite images used for the investigation.**

| Acquisition Date | Path/Row | Landsat | Sensor | Spatial Resolution | Number of Bands |
|---|---|---|---|---|---|
| 10/01/1991 | 193/056 | Landsat 4 | TM | 30m | 7 |
| 26/12/2002 | 193/056 | Landsat 7 | ETM+ | 30m | 9 |
| 06/12/2015 | 193/056 | Landsat 8 | OLI-TIRS | 30m | 11 |

**Source**: Fieldwork, 2020

## 2.2 Data sources

The main data for this study were Landsat images and flood disaster statistics. A spatial resolution of 30m Landsat satellite surface reflectance images of 1991, 2002 and 2015 were acquired from the United States Geological Survey (USGS) Earth Explorer site (https://glovis.usgs.gov). These images were extracted to assess temporal and spatial changes in the study area. Landsat TM, ETM+ and Landsat 8 images were considered (see Table 1) due to their high resolutions which would enable changes of urban areas from other land use/land cover surfaces to be detected. These periods were chosen because the negative impacts of urbanization in Accra such as flood disasters became serious from the 1990s and these impacts are a major challenge for developers and other stakeholders in the area. Furthermore, existing literature already indicates that the land area of Accra built up in periods before the year 2015 [7]. Additionally, the readily availability of data for flood data disasters in the study site for the period is also the reason for choosing those periods. Data on flood risk zones and flood disaster events were obtained from secondary sources and Ghana's National Disaster Management Organization (NADMO) archives.

## 2.3 Pre-process of Landsat images

Top-of-atmosphere (TOA) calibration was used to convert the Digital Number (DN) values to reflectance in order to minimize the radiometric differences caused by disparities in TM and Landsat 8 sensors as a result of sensor-target-illumination. Geometrically, the images were corrected and calibrated to reflectance by means of information provided in the image metadata. Atmospherically calibrated images were mosaicked and clipped to the extent of the study area boundary.

## 2.4 Classification and accuracy assessment

Hybrid approach to image classification was used to classify the images with the aid of GIS software and ERDAS imagine. This involves conducting unsupervised classification first by defining signature files and fixing the number of classes to undertake preliminary interpretation. The main benefit of employing the unsupervised classification is the identification of distinct spectral classes for the images. It was then followed by supervised classification techniques by using the maximum likelihood classification algorithm, where a comparison of the spectral properties of each pixel were made to a set of representative pixels, which are known as training data or sample specified by the user. The training samples were collected using the researcher's personal knowledge and experience of the study area, as well as Google Earth and topographic maps. In all, the satellite images were classified into various classes of urban, grassland, water, bareland, forest, agriculture and shrublands (Table 2). These classes were identified and grouped in this study based on the USGS Land use/land cover classification scheme developed by [28]. However, the classification system was fairly improved to suit the study area.

**Table 2. Description of land use/land cover types.**

| Land use/land cover categories | Description |
|---|---|
| Urban | Mixed urban areas, Residential areas, Industrial and Commercial units, Transportation facilities |
| Grassland | Mixed grassland with few scattered tree |
| Water, | Lakes, Rivers, Streams, Canals, Reservoirs, Estuaries, Bays, Forested wetlands, Aquaculture facilities |
| Bareland, | Beaches, Sandy areas, Exposed rocks, River banks, Quarries and gravel pits, transitional areas |
| Agriculture | Pasture, cropland, orchards and fallow land |
| Shrubland | Herbaceous vegetation, Shrub and bush areas, |
| Forest | Evergreen forest land, Mixed forest land, Deciduous forest land, Forest reserves |

**Source**: Fieldwork, 2020

Confusion matrix was used to assess the accuracy of the classified images. This includes the omission error, commission error and kappa statistic. Ground-truth data obtained from GPS field surveys, Landsat images, existing maps and Google earth images were used as reference data. A minimum of sixty samples for each class was taken based on the rule-of-thumb theory recommended by [29].

The post-classification change detection technique was adopted to quantify, identify, and describe the differences in the land use maps produced. This technique is capable of generating a complex matrix of change and also minimizing the effects of sensor and atmospheric differences between two dates by using classified images [30].

## 2.5 Land use/land cover change detection

Change detection is the procedure of distinguishing the differences in the settings of both natural and man-made features on earth at different periods [31, 32]. Several approaches can attain change detection analysis by detecting and measuring differences in the image acquired at different time points over the same area. In this study, change detection was attained through transition matrices. This was achieved through "subtraction" of any two classified images via a pixel-by-pixel comparison. This generates a matrix with diagonal representing area statistics of categories that have not changed between the time points compared. The non-diagonal cells represent the dynamics transition between the initial period and the final period.

## 2.6 Flood risk mapping

Data on topography, hydrology, land use and historical flood events are used to map flood risk. To ensure consistency, the collected data was cleaned and formatted. ArcGIS Pro was used to analyze the clean data and identify areas of potential flood risk. This entails creating digital elevation models, calculating flow accumulation and direction, and identifying areas of potential flooding based on historical flood events and land use/land cover of the study area. The analyzed data was then used to perform a risk assessment to determine the level of flood risk in different zones within the study area. The assessment considered the study area's topography, hydrology, and land use/land cover. Finally, a flood risk map was created using ArcGIS Pro. This consists of creating a base map, overlaying the analyzed data, and symbolizing the data to show the levels of flood risk.

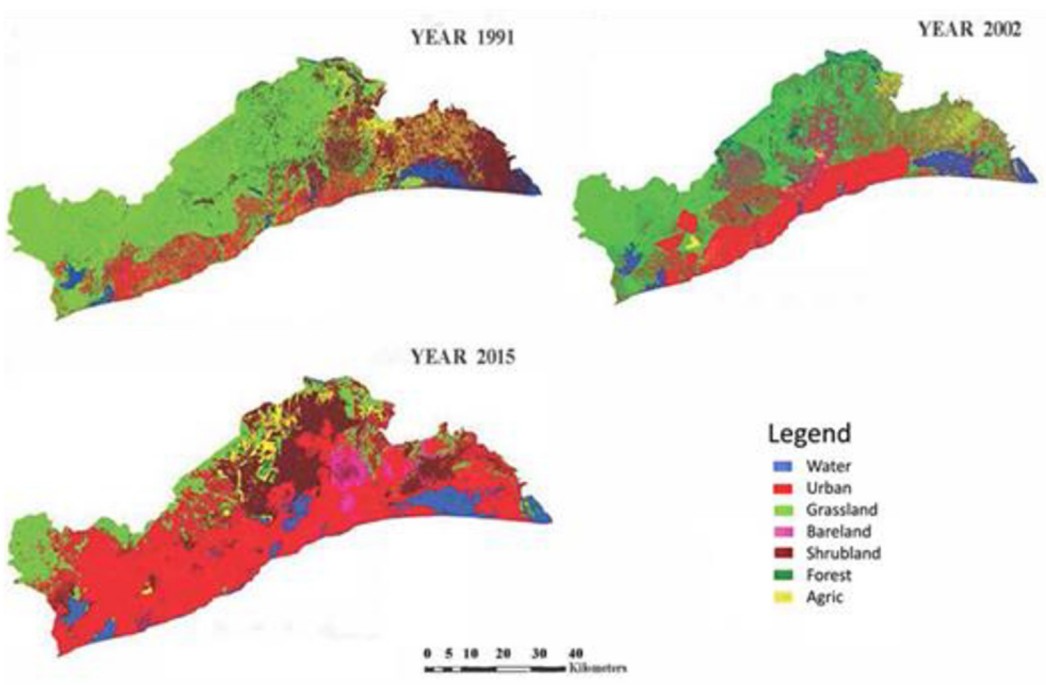

**Fig 2. Land use/land cover distribution in three time points. Source**: Fieldwork, 2020.

## 3 Results and discussion

### 3.1 Spatial distribution of land use/land cover of the study area

Based on the classification of the satellite image data of the three periods in the study area, the spatial distribution map of land use/land cover in different time points was generated (Fig 2). Statistics on land use/land cover types in the corresponding years (Table 3) were also produced to analyze the quantitative and spatial characteristics of land use changes.

**Table 3. Summary of land use/land cover type structure of the study area.**

| Land use type | | 1991 | 2002 | 2015 |
|---|---|---|---|---|
| Water | Area(km$^2$) | 185.974 | 203.914 | 236.865 |
| | Proportion (%) | 5.09 | 5.6 | 6.51 |
| Urban | Area(km$^2$) | 403.51 | 788.894 | 1807.69 |
| | Proportion (%) | 11.09 | 21.68 | 49.67 |
| Grassland | Area(km$^2$) | 1887.72 | 1475.81 | 497.312 |
| | Proportion (%) | 51.9 | 40.55 | 13.67 |
| Bare land | Area(km$^2$) | 32.5791 | 90.022 | 169.179 |
| | Proportion (%) | 0.9 | 2.47 | 4.65 |
| Shrubland | Area(km$^2$) | 868.95 | 887.063 | 773.807 |
| | Proportion (%) | 23.87 | 24.37 | 21.26 |
| Forest | Area(km$^2$) | 11.680 | 63.684 | 31.086 |
| | Proportion (%) | 0.32 | 1.75 | 0.85 |
| Agric | Area(km$^2$) | 248.846 | 129.872 | 123.33 |
| | Proportion (%) | 6.84 | 3.57 | 3.39 |

**Source**: Fieldwork, 2020

Fig 2 shows that the main land use/land cover classes were grassland, shrubland and urban. Grassland was the largest land cover, mainly distributed in the south western, central, nothern and north eastern plains of the study area in 1991. The area of grassland accounts for over 50% of the total area of the study area, but keep on decreasing gradually from about 51% in 1991 to 13% in 2015. This shows a decrease of more than hundred percent, and significant decline in the area of grassland. This can be to the rapid expansion of urban areas at the expense of grassland and agriculture areas. The urban area is mainly distributed within the southern, south west, along the central to south eastern, towards the northern part of the study area. From 1991 to 2015 urban area has increased significantly more than hundred percent, which can be attributed to population increase and the need for accomodation by the rapidly growing population in the study area. Though shrubland cover is second and larger than urban, forest, water and agric (agriculture) classes, urban area in 1991 stands as the third largest land use type in the study area. From Fig 1, it also is obvious that, in 2002, grassland occupies the largest area although it has reduced in coverage relative to 1991. The urban class has increased extremely to more than hundred percent from 1991 to 2002 and this can be explained by the possible factors of urbanization mentioned earlier. Also, drastic changes in the land use types were observed in 2015. Apparently, the urban class has increased to be the largest land surface and has eaten up most of the grassland, agriculture land and forest cover surface of the study area. The forest land cover has reduced to be the smallest land cover of the region and water land cover seems to be increasing. This is similar to the assertion of [4] that the urban area has increased, and it is projected to increase to over 50% between 2015 and 2025.

## 3.2 Nature, extent and rate of land/use change

The statistics revealed that, in 1991, the main land-cover was grassland and shrubland, occupying about 51.9% and 23.9% of the total area respectively. The results further indicate that the third major land cover was urban area with 11.1% of the total land use/land cover of the study site. Agriculture area occupied 6.9% as at 1991, while water, bare land and forest occupies 5.1%, 0.9% and 0.3% of the total land use/land cover area respectively. It was observed that urban areas in the region have drastically expanded, mainly at the expense of grassland and agriculture areas. Forest areas, grassland and agriculture in the region had the greatest loss, while shrubland and water bodies showed very little change between 1991 and 2015. During 2002, different distributions of the land use/land cover classes were identified compared to 1991. Urban area increased from 1991 to 2002 to cover 21.7% of the total land use/ land cover area. The analysis also indicated that urban areas had significantly increased, expanding from 11.1% in 1991 to 21.7% in 2002. This finding agrees with [4]. This percentage increase means that urban covered approximately 788.894km$^2$ of the region in 2002 Between the same time period, grassland and agriculture area reduced to 40.6% and 3.6% respectively. Table 4 shows the land-use percentage and area changes for the two study periods. Urban areas continued to massively expand and became the main land use type in the region in 2015. Thus, it covered almost half of the total land use/land cover of the study area, amounting to 49.7% (1807.69km$^2$) in the year. Grassland reduced extremely within the same time period, occupying only 13.7% of the study area in 2015. Fig 3 shows the percentage coverage of the land-use types for the years 1991, 2002 and 2015. Table 4 also presents the land-use percentage and area changes for the two study periods. The study observed a drastic continued increased in urban areas, by 95.51% and 129.14% in the periods 1991–2002 and 2002–2015 respectively.

**Table 4. Land-use percentage change and rate of change during the two study periods.**

| LULC types | Percentage change (%) | | Annual rate of change | | Area change (km sq.) | |
|---|---|---|---|---|---|---|
| | 1991–2002 | 2002–2015 | 1991–2002 | 2002–2015 | 1991–2002 | 2002–2015 |
| Water | 9.65 | 16.16 | 0.88 | 1.24 | 17.94 | 32.95 |
| Urban | 95.51 | 129.14 | 8.68 | 9.93 | 385.38 | 1018.80 |
| Grassland | -21.82 | -66.30 | -1.98 | -5.1 | -411.91 | -978.50 |
| Bareland | 176.32 | 87.93 | 16.03 | 6.76 | 57.44 | 79.16 |
| Shrubland | 2.08 | -12.77 | 0.19 | -0.98 | 18.11 | -113.26 |
| Forest | 445.23 | -51.19 | 40.48 | -3.94 | 52.00 | -32.60 |
| Agric | -47.81 | -5.04 | -4.35 | -0.39 | -118.97 | -6.54 |

**Source**: Fieldwork, 2020

### 3.3 Change detection analysis

Change detection statistics were performed to obtain the land use/land cover changes for each of the intervals: 1991–2002 and 2002–2015. The results are shown in Tables 5 and 6. Table 5 indicates that, out of the 1887.72 km$^2$ that was grassland in 1991, 995.69 km$^2$ was still grassland area in 1991 and the rest converted to urban and shrubland. The reduction in grassland during the period 1991–2002 was mainly due to the expansion of urban area and shrubland. Grassland area was converted into urban area of about 283.92 km$^2$ accounting for about 15% of the grassland area in 1991. Urban has gained from the area of shrubland of about 149.03km$^2$ representing about 17% of the shrubland in 1991. The main land use types converted into urban area during the period were grassland, shrubland and agriculture. Similarly, from 2002 to

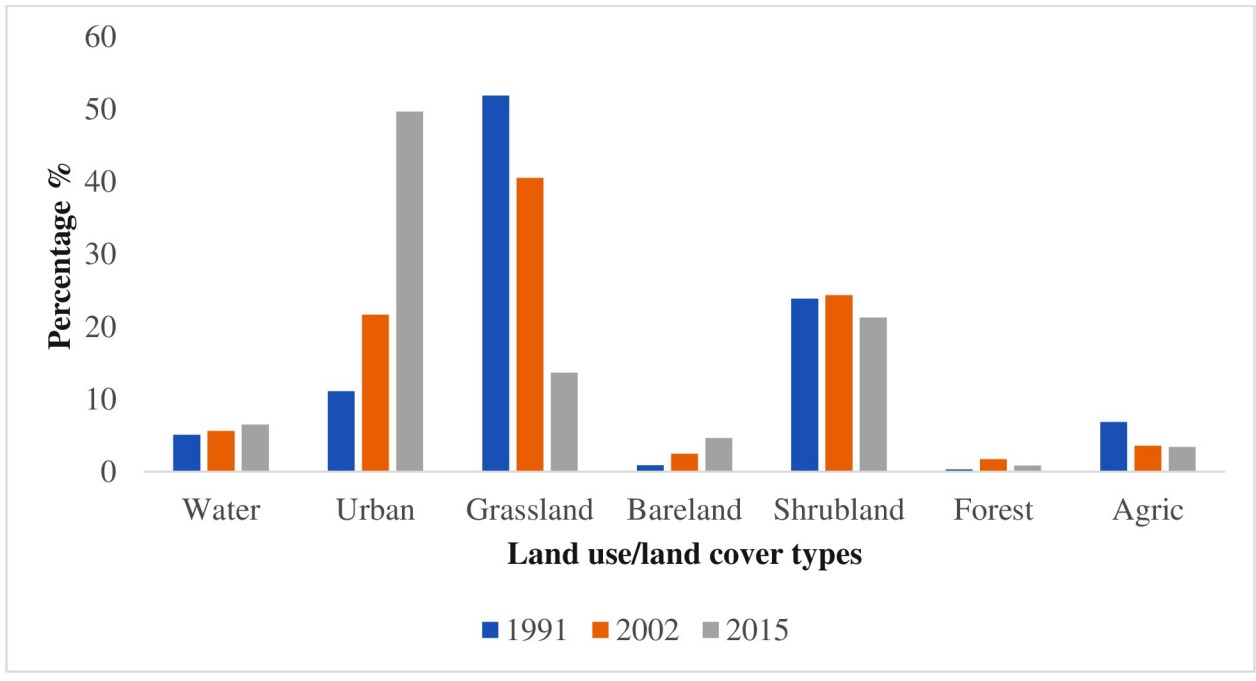

**Fig 3. Land-use type percentage coverage for 1991, 2002, and 2015. Source**: Fieldwork, 2020.

**Table 5. Change detection matrix of land use in the study area from 1991 to 2002, unit (km$^2$).**

|  | Water | Urban | Grassland | Bareland | Shrubland | Forest | Agric | Total 2002 |
|---|---|---|---|---|---|---|---|---|
| Water | 118.96 | 2.50 | 41.32 | 3.99 | 30.64 | 2.85 | 3.66 | 203.91 |
| Urban | 34.57 | 276.15 | 283.92 | 9.60 | 149.03 | 0.63 | 35.00 | 788.89 |
| Grassland | 14.50 | 54.64 | 995.69 | 8.36 | 307.29 | 3.25 | 92.10 | 1475.81 |
| Bareland | 8.79 | 18.32 | 30.09 | 4.41 | 22.21 | 0.03 | 6.17 | 90.02 |
| Shrubland | 6.76 | 43.92 | 471.11 | 5.77 | 280.96 | 0.76 | 77.78 | 887.06 |
| Forest | 0.51 | 0.43 | 39.58 | 0.05 | 12.88 | 3.95 | 6.28 | 63.68 |
| Agric | 1.89 | 7.56 | 26.00 | 0.41 | 65.95 | 0.21 | 27.86 | 129.87 |
| **Total 1991** | 185.97 | 403.51 | 1887.72 | 32.58 | 868.95 | 11.68 | 248.85 | 3639.26 |

**Source**: Fieldwork, 2020

**Table 6. Change detection matrix of land use in the study area from 2002 to 2015, unit (km$^2$).**

|  | Water | Urban | Grassland | Bareland | Shrubland | Forest | Agric | Total 2015 |
|---|---|---|---|---|---|---|---|---|
| Water | 143.84 | 37.32 | 17.64 | 5.72 | 28.51 | 0.93 | 2.89 | 236.86 |
| Urban | 13.09 | 656.02 | 636.94 | 59.06 | 369.12 | 11.79 | 61.66 | 1807.69 |
| Grassland | 21.52 | 9.94 | 334.50 | 2.38 | 88.81 | 23.35 | 16.81 | 497.31 |
| Bareland | 0.71 | 38.57 | 66.56 | 6.60 | 49.74 | 0.82 | 6.19 | 169.18 |
| Shrubland | 11.16 | 46.69 | 344.04 | 15.83 | 315.65 | 7.33 | 33.12 | 773.81 |
| Forest | 4.81 | 0.05 | 10.61 | 0.04 | 3.44 | 10.17 | 1.96 | 31.09 |
| Agric | 8.77 | 0.32 | 65.51 | 0.40 | 31.79 | 9.30 | 7.24 | 123.33 |
| **Total 2002** | 203.91 | 788.89 | 1475.81 | 90.02 | 887.06 | 63.68 | 129.87 | 3639.26 |

**Source**: Fieldwork, 2020

2015, grassland, shrubland and agriculture areas continued to be converted into urban area as shown in Table 6.

### 3.4 Flood risk zones within the study area

Based on the topography, hydrology, land use/land cover and historical flood events of the study area, the flood risk zones of the area and the levels of risk were generated (Fig 4).

### 3.5 The trends of urbanization and frequency of flood disaster events in the study areas

The land use/land cover change analysis shows that urban areas keep increasing at the expense of forest and grassland which cause flooding in the area. For instance, urban areas increased from 11.1% in 1991 to 21.7% in 2002. This is about double in urban expansion within a period of 10 years. Whereas grassland was reduced by 11.4%, there was insignificant increment of 1.3% in forest land cover of the area during the same period. Besides, from 2002 to 2015, grassland, shrubland and agricultural lands continue to urban areas (Table 6). Coupled with these changes and unplanned development of physical structures which encroach on waterways, the risk of flooding in Accra is increased when there is rainstorm. Studies such as [3, 7, 33] confirmed issues of urbanization and floods as major urban problems in Greater Accra.

Table 7 shows the trajectory of flood events from 1990 to 2015 in the study area. The results show that 8 and 11 flood events occurred in the periods of 1991 to 2002 and 2002 to 2015

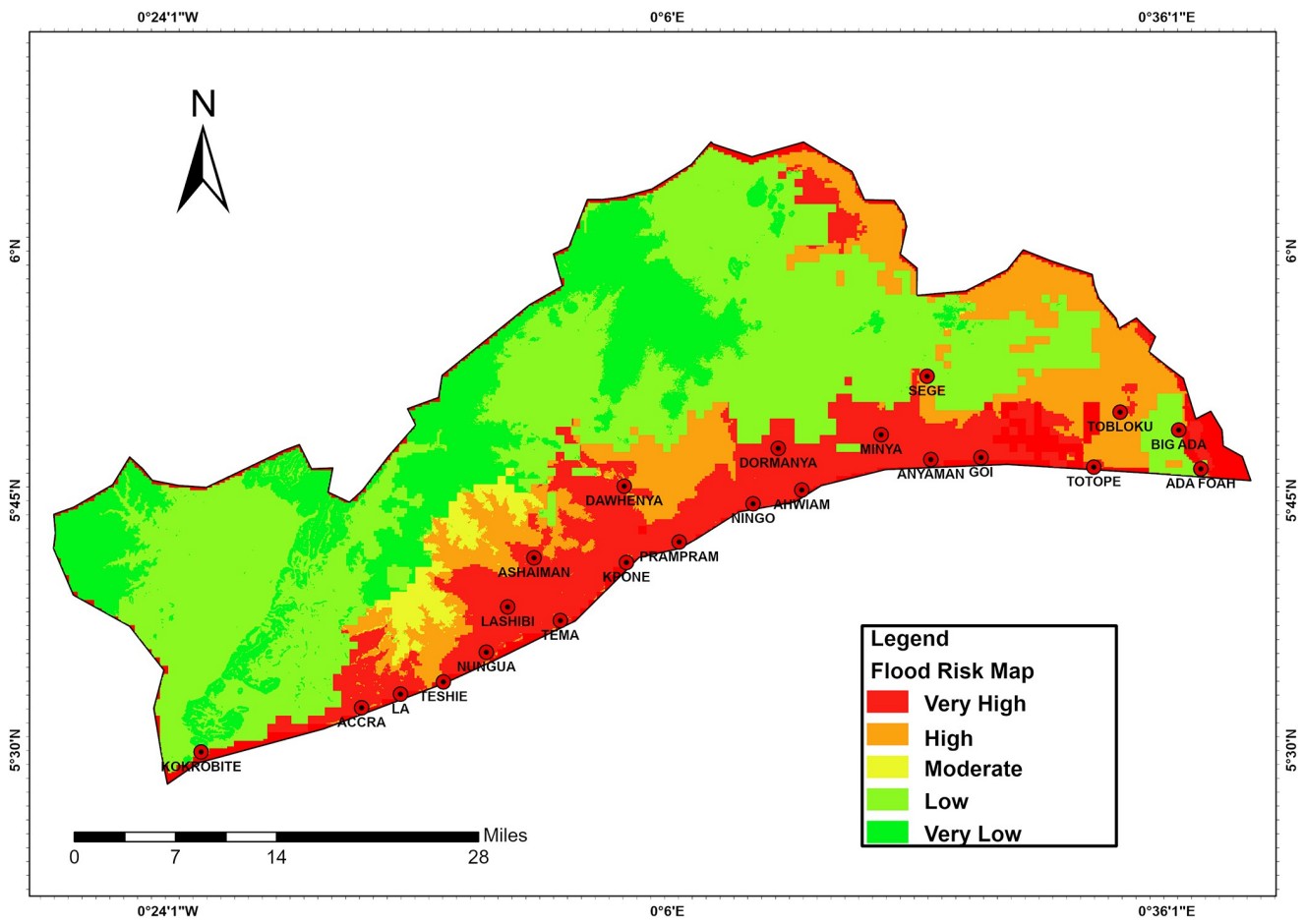

**Fig 4. Flood risk zones and levels of the area. Source**: Fieldwork, 2020.

**Table 7. Historical records of floods since 1955 and frequency of flood events between 1991 to 2002 and 2002 to 2015.**

| 1990–2002 Period | 2003–2015 Period |
|---|---|
| **Flood events** | **Flood events** |
| 27 Nov. 1990 | 26-Mar 2007 |
| 15 July 1991 | 03-June 2007 |
| 18 Nov 1993 | 18-May 2008 |
| 5–6 June 1994 | 19 June 2009 |
| 04-July 1995 | 5 May 2010 |
| 13 June 1997 | 24-Feb 2011 |
| 28-June 2001 | 01-Nov 2011 |
| 09 June 2002 | 31-May 2013 |
| | 06-June 2014 |
| | 04-July 2014 |
| | 03-Jun 2015 |
| Total: 8 flood events | Total: 11 flood events |

**Sources**: NADMO Disaster Reports.

respectively. It is clear in the two periods that there is reoccurrence of flood events for the period of 2002 to 2015 and annual flood incidence is also consistent from 2007 up to 2015 whereas the period 1991 to 2002 does not have this consistency in the flood events. In 2011 and 2014 for instance, flood events reoccurred twice in each year. The patterns of increase in reoccurrence of floods seem to correlate with the sharp increase in urban areas culminating in reduction in grassland and forest cover of the 2002 to 2015 period. Other factors such as increase in rainfall and sea level rise contribute to flood risk in Accra [34]. However, literature further contends that flood risk in the study site has increased in recent years as a result of unregulated urbanization [3]. Notwithstanding this, rainfall records since 1960 till date have not increased and could not be a major factor for flood disaster risk [21, 33]. [3] further show that urbanization has increased upward since the 2000s and encroachment on waterways has been repeatedly reported as a reason for increase in flood disaster risk Accra [7, 34].

This researched revealed that earlier studies have already mapped the flood risk zones in Accra as the centre of the study. [21] mapped out flood risk zones in the city of Accra using GIS and confirmed that medium to high flood risk zones constituted to about 40% of flood zones. However, our studies showed that urbanization has increased beyond those zones that were marked by those studies conducted earlier, and the frequency of flood occurrence has also increased. This correspondence means that more areas have been urbanized in an unplanned manner and more areas have become vulnerable to flood hazards thereby increasing the risk of flooding in the study site. Analysis of our research shows that the epicentre of flooding in the study area is the AMA which previous studies on flood risk revealed that urbanization with its consequences of encroachment on waterways results in the increase in flood occurrence [7]. From our analysis and with increase in urban expansion beyond the AMA, it can be envisaged that encroachment on waterways through unplanned development has increased and this suggests a positive correlation with the corresponding increase in frequency of flood occurrence in the study site.

## 4 Conclusions, limitations and recommendations

The findings revealed that the trend of urbanization increased in the Greater Accra Region for period of 1991 to 2015 and this corresponds with the increase in the frequency of flood events. The study mapped and analyzed urbanization of land use/land cover change in Accra for the periods of 1991–2002 and 2002–2015, using geospatial analytical techniques. Of the various land use land cover classes that were mapped and analyzed, it can be deduced that urban land cover area did not cover a large area in 1991 but it increased to become the largest land use/land cover surface of the study site in the year 2015. Out of historical flood events since 1955, between 1991 and 2015, flood disaster events were identified which indicated an increased in the pattern of flood occurrence in the study site. The analysis of the relationship between the pattern of urbanization and frequency of flood disaster events in the study area between 1991 and 2015 suggests that there is both an increase in urban land surface and the frequency of flood events. It was also observed that the pattern of urbanization in the study site took place in an unplanned style, where physical developments in waterways became high. This unplanned expansion in urban development resulted in encroachment on waterways and flood risk zone areas. The findings therefore suggest that the increase in urbanization corresponds with the increase in the frequency of flood disaster events in the study area from 1991 to 2015.

The main limitations of this study include the following. Due to this limitation, it is clear how much percentage is rate of urban land use land cover change. A future research should take up a a comprehensive quantitative analysis of these two processes of urbanization and

flood disaster risk management can be considered in future studies using remote sensing and GIS. For urban planning and development policy, it is recommended that urbanization and flood disaster risk are a major environmental problem of the study site and need to be tackled. It is recommended land use policy decision makers should ensure that developers and stakeholders abide by the land use policies and regulations of the study area. This will ensure the growing expansion of urban areas can be minimized to reduce flood disaster risk in the study area.

## Supporting information

**S1 File. Description of supplementary material file.**
(DOCX)

## Acknowledgments

We acknowledge Centre for Remote Sensing and Geographic Information Services, University of Ghana, Legon for providing us information for this research.

## Author Contributions

**Conceptualization:** Raphael Ane Atanga, Vitus Tankpa.

**Data curation:** Raphael Ane Atanga, Vitus Tankpa, Isaiah Acquah.

**Formal analysis:** Raphael Ane Atanga, Vitus Tankpa.

**Investigation:** Raphael Ane Atanga, Vitus Tankpa.

**Methodology:** Vitus Tankpa.

**Project administration:** Raphael Ane Atanga, Vitus Tankpa.

**Resources:** Vitus Tankpa.

**Software:** Vitus Tankpa.

**Supervision:** Vitus Tankpa.

**Validation:** Vitus Tankpa.

**Visualization:** Vitus Tankpa.

**Writing – original draft:** Vitus Tankpa.

**Writing – review & editing:** Vitus Tankpa.

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
