## [Decision Letter · Decision Letter 0]

21 Jun 2023

PONE-D-23-15951“URBANIZATION AND FLOOD RISK ANALYSIS USING GEOSPATIAL TECHNIQUESPLOS ONE

Dear Dr. Tankpa,

Thank you for submitting your manuscript to PLOS ONE. After careful consideration, we feel that it has merit but does not fully meet PLOS ONE’s publication criteria as it currently stands. Therefore, we invite you to submit a revised version of the manuscript that addresses the points raised during the review process. The anonymised comments to authors, from all reviewers, are included below and in attachments.

We look forward to receiving your revised manuscript.

Kind regards,

Mohammed Sarfaraz Gani Adnan, PhD

Academic Editor

PLOS ONE

Journal Requirements:

6. We note that Figures 1,2 and 4 in your submission contain [map/satellite] images which may be copyrighted. All PLOS content is published under the Creative Commons Attribution License (CC BY 4.0), which means that the manuscript, images, and Supporting Information files will be freely available online, and any third party is permitted to access, download, copy, distribute, and use these materials in any way, even commercially, with proper attribution. For these reasons, we cannot publish previously copyrighted maps or satellite images created using proprietary data, such as Google software (Google Maps, Street View, and Earth). For more information, see our copyright guidelines: http://journals.plos.org/plosone/s/licenses-and-copyright.

a. You may seek permission from the original copyright holder of Figures 1,2 and 4 to publish the content specifically under the CC BY 4.0 license.  

Reviewers' comments:

Reviewer's Responses to Questions

**Comments to the Author**

1. Is the manuscript technically sound, and do the data support the conclusions?

Reviewer #1: Partly

Reviewer #2: Partly

2. Has the statistical analysis been performed appropriately and rigorously? 

Reviewer #1: Yes

Reviewer #2: Yes

3. Have the authors made all data underlying the findings in their manuscript fully available?

Reviewer #1: No

Reviewer #2: Yes

4. Is the manuscript presented in an intelligible fashion and written in standard English?

Reviewer #1: No

Reviewer #2: Yes

5. Review Comments to the Author

Reviewer #1: This manuscript presented Urbanization and Flood Risk Analysis using Geospatial Techniques. The overall quality of this paper is worthy of publication. The authors could reply to general and scientific issues and revise their papers accordingly.

1. Is the author’s focus on urban/pluvial flood here? In the current version, it is pretty tricky to understand the type of flood. I’d suggest stressing this in the revision.

2. The abstract is not well written. Please emphasize the following contents: background, objectives, methods, results, and concluding remarks.

3. Research questions were not correctly indicated. What are the problems solved by the authors in their existing investigation? Please stress it. See these papers for your reference:

https://link.springer.com/article/10.1007/s11069-022-05357-0

https://www.sciencedirect.com/science/article/pii/S1674987122000780

4. The choice of the model should be justified.

5. Why does the scientific community cite this work? There is nothing observed regarding the new contribution of the findings other than selecting the case study.

6. Literature review is poorly done. The latest work on flood modeling should be well justified. Therefore, the literature review needs to be well-expanded.

7. The selection of the study area and data used in this paper need more justification. Like, Landsat 4 TM data has scale line errors in many cases. If the author faced the same issue, how did they solve it?

8. “To ensure consistency, the collected data was cleaned and formatted.” How? Need justification.

9. Have the authors experimented with additional classification schemes for constructing a flood risk map? Currently, they assigned the five classes map. Have they assessed the impact of using additional classes? If so, do the spatial distribution of the different classes in the flood risk map and the derived results change significantly?

10. The authors should also clearly explain how they classified their flood risk map into five groups. Changing the limits of the zones can alter the entire result of their study, so a justification is needed.

11. Finally, the authors should enhance the discussion section by discussing: (i) the implications of their findings in the context of the current trend (and necessity) to evaluate flood hazards and secondary effects of a specific phenomenon. (ii) can the approach of the present paper have practical usage for planners, and how can their findings be communicated? In other words, the authors should emphasize the added value of their paper in the context of the topic.

Reviewer #2: The research focuses on the analysis of urbanization and flood risk in the Greater Accra Region of Ghana. The study aims to map and analyze the trends of urbanization, identify flood risk zones, and analyze the relationship between urbanization patterns and flood disaster events from 1990 to 2015. The research also aims to contribute to the understanding of urbanization and flood disaster risk in the study area and similar regions. It emphasizes the importance of effective urban planning, land use policies, and regulations to minimize the expansion of urban areas and reduce flood disaster risk. I enjoyed reading the paper. However, I have some comments for further improvement of the paper. I suggest major revision

6. PLOS authors have the option to publish the peer review history of their article (what does this mean?). If published, this will include your full peer review and any attached files.

Reviewer #1: **Yes: **Dr. Md. Mahfuzur Rahman

Reviewer #2: No

---

## [Author Response · Author response to Decision Letter 0]

2 Sep 2023

Reviewer #1’s comments: 

This manuscript presented Urbanization and Flood Risk Analysis using Geospatial Techniques. The overall quality of this paper is worthy of publication. The authors could reply to general and scientific issues and revise their papers accordingly.

1. Is the author’s focus on urban/pluvial flood here? In the current version, it is pretty tricky to understand the type of flood. I’d suggest stressing this in the revision.

2. The abstract is not well written. Please emphasize the following contents: background, objectives, methods, results, and concluding remarks.

3. Research questions were not correctly indicated. What are the problems solved by the authors in their existing investigation? Please stress it. See these papers for your reference:

https://link.springer.com/article/10.1007/s11069-022-05357-0

https://www.sciencedirect.com/science/article/pii/S1674987122000780

4. The choice of the model should be justified.

5. Why does the scientific community cite this work? There is nothing observed regarding the new contribution of the findings other than selecting the case study.

6. Literature review is poorly done. The latest work on flood modeling should be well justified. Therefore, the literature review needs to be well-expanded.

7. The selection of the study area and data used in this paper need more justification. Like, Landsat 4 TM data has scale line errors in many cases. If the author faced the same issue, how did they solve it?

8. “To ensure consistency, the collected data was cleaned and formatted.” How? Need justification.

9. Have the authors experimented with additional classification schemes for constructing a flood risk map? Currently, they assigned the five classes map. Have they assessed the impact of using additional classes? If so, do the spatial distribution of the different classes in the flood risk map and the derived results change significantly?

10. The authors should also clearly explain how they classified their flood risk map into five groups. Changing the limits of the zones can alter the entire result of their study, so a justification is needed.

11. Finally, the authors should enhance the discussion section by discussing: (i) the implications of their findings in the context of the current trend (and necessity) to evaluate flood hazards and secondary effects of a specific phenomenon. (ii) can the approach of the present paper have practical usage for planners, and how can their findings be communicated? In other words, the authors should emphasize the added value of their paper in the context of the topic

Reviewer #2 comments: 

The research focuses on the analysis of urbanization and flood risk in the Greater Accra Region of Ghana. The study aims to map and analyze the trends of urbanization, identify flood risk zones, and analyze the relationship between urbanization patterns and flood disaster events from 1990 to 2015. The research also aims to contribute to the understanding of urbanization and flood disaster risk in the study area and similar regions. It emphasizes the importance of effective urban planning, land use policies, and regulations to minimize the expansion of urban areas and reduce flood disaster risk. I enjoyed reading the paper. However, I have some comments for further improvement of the paper. I suggest major revision

RESPONSE: Thank you very much for all the valuable comments by the reviewer

Response to Reviewer #1’s comments:

1. Is the author’s focus on urban/pluvial flood here? In the current version, it is pretty tricky to understand the type of flood. I’d suggest stressing this in the revision. 

RESPONSE: 

Flash floods are the common major problem in the Greater Accra, influenced by severe localized storms and that is the focus of this research.

2. The abstract is not well written. Please emphasize the following contents: background, objectives, methods, results, and concluding remarks.

RESPONSE: 

Contemporarily, flood disasters are a critical global problem. More especially, urban floods are a developmental problem for metropolitan areas in the global south and the flood disasters in Accra are illustrative case. However, literature on geospatial analysis of the relationship between urbanization and flood disasters in Accra appears unclear. Data on flood disasters and urbanization were obtained from existing sources and analyzed using geospatial techniques. The main data for this study were Landsat images and flood disaster statistics. Three cloud-free spatial resolution of 30m Landsat satellite surface reflectance images of 1991, 2002 and 2015 were acquired from the United States Geological Survey (USGS) Earth Explorer.

3. Research questions were not correctly indicated. What are the problems solved by the authors in their existing investigation? Please stress it. See these papers for your reference:

https://link.springer.com/article/10.1007/s11069-022-05357-0

https://www.sciencedirect.com/science/article/pii/S1674987122000780

Response: The main focused on addressing the specific research objectives (see lines 169 -176).

This research has three specific research objectives. The first objective is to map and analyze the trends of urbanization as a land use land type in Greater Accra for the periods of 1991 -2002 and 2002 – 2015 using geospatial techniques. The second objective is to map flood risk zones, identify the flood disaster events between 1990 and 2015 and analyze the trends of flood disaster risk in Greater Accra. The third objective is to analyze the relationship between the pattern of urbanization and the increase in flood disaster events from 1990 to 2015. The aim of this research is to contribute to research, policy and practical management of urbanization and flood disaster risk in the study site and other areas with similar conditions. 

4. The choice of the model should be justified.

RESPONSE: Myriad of studies investigated the menace of flooding in Accra without specific emphasis on the relationship between urbanization and flood disasters using geospatial techniques. Research about such relationship is useful to inform disaster risk managers and urban development planners for better flood disaster risk reduction. Seemingly, the literature indicates flooding as a big problem in Greater Accra as well as its causes, impacts and management challenges (Karley, 2009; Rain et al., 2011; Jha et al., 2012; Amoako and Boamah, 2015; Douglas, 2017). The literature further attributes the floods in Accra to the impacts of climate change (Rain et al., 2011). The flooding Accra is typically pluvial flooding caused by localized rains, poor and inadequate drainage, physical development on waters among others (Karley, 2009; Okyere et al., 2013). Studies applied remote sensing and GIS techniques to investigate the issue of flooding in Accra but did not specifically focus the flood disaster risk and urbanization relationships (Nyarko, 2002; Konadu and Fosu, 2009; Jankowska et al., 2011; Owusu et al., 2013; Yeboah et al., 2017). 

5. Why does the scientific community cite this work? There is nothing observed regarding the new contribution of the findings other than selecting the case study.

RESPONSE: Contemporarily, flood disasters are a critical global problem. More especially, floods are a developmental problem for urban regions in the global south and the flood disasters in the Greater Accra in Ghana are an illustrative case. However, literature on geospatial analysis of the relationship between urbanization and flood disasters in the Greater Accra appears unclear. The aim of this research will to contribute to research, policy and practical management of urbanization and flood disaster risk in the study site and other areas with similar conditions. Understanding the relationship between urbanization as a land use/land cover change and the occurrence of flood disasters in Great Accra would provide evidence for urban development planners, policy makers and flood managers to coordinate for effective response to the menace of urbanization and flood disaster risk. This research will further to literature and academic discussion in the areas of geospatial analysis, urban flood disaster risk management and urban geography in general. 

6. Literature review is poorly done. The latest work on flood modeling should be well justified. Therefore, the literature review needs to be well-expanded.

RESPONSE: Seemingly, the literature indicates flooding as a big problem in Greater Accra as well as its causes, impacts and management challenges (Karley, 2009; Rain et al., 2011; Jha et al., 2012; Amoako and Boamah, 2015; Douglas, 2017). The literature further attributes the floods in Accra to the impacts of climate change (Rain et al., 2011). The flooding Accra is typically pluvial flooding caused by localized rains, poor and inadequate drainage, physical development on waters among others (Karley, 2009; Okyere et al., 2013). Studies applied remote sensing and GIS techniques to investigate the issue of flooding in Accra but did not specifically focused the flood disaster risk and urbanization relationships (Nyarko, 2002; Konadu and Fosu, 2009; Jankowska et al., 2011; Owusu et al., 2013; Yeboah et al., 2017). Asumadu-Sarkodie, Owusu and Rufangura (2015) analyzed the causes of flood in Accra and the suitable structural measures for mitigating flood impacts. The floods usual from emanated several hours of localized rainfall. The authors further revealed that an integrated flood risk management approach to dealing with flood impacts. The monthly precipitation in Accra was projected increase from 160 mm in 1991-2010 to 200 mm in 2011-2020. Ullah et al. (2022) further argued that an effective multi-hazard risk mitigation strategy would require spatiotemporal analysis individual hazards and their interactions using geospatial techniques. The authors proposed a multi-hazard susceptibility mapping framework using the classical deep learning algorithm of Convolutional Neural Networks. Nkonu et al. (2023) used GIS-based model for flood vulnerability analysis in the Accra Metropolitan Area. The analysis revealed that areas with high flood vulnerability increased between 2007 and 2020, and the high flood vulnerability zones are largely located in the south-central and south-western regions. Ackom, Adjei and Odai (2020) assessed the extent and trends of changes in the urban environment of Odaw River Basin from 1991 to 2016 in Accra, which revealed changes in rates of land degradation during the period with over 200% upsurge in settlement. Sajjad, Lu, Aslam, and Ahmad (2023) applied geospatial techniques to model flood extent as well as flood duration in District Dera Ghazi Khan, Pakistan. The results showed that the flood lasted for nearly 5 weeks in the study area. Results enabled identification of inundated areas to allow for emergency responses to newly flooded areas.

7. The selection of the study area and data used in this paper need more justification. Like, Landsat 4 TM data has scale line errors in many cases. If the author faced the same issue, how did they solve it?

RESPONSE: The study area was selected for the research due to its increasing urbanization and protracted annual flood disasters in recent years. The main data for this study were Landsat images and flood disaster statistics. Three cloud-free spatial resolution of 30m Landsat satellite surface reflectance images of 1991, 2002 and 2015 were acquired from the United States Geological Survey (USGS) Earth Explorer site (HTTP:\\\\glovis.usgs.gov). These images were extracted to assess temporal and spatial changes in the study area. Landsat TM, ETM+ and Landsat 8 images were considered due to their high resolutions which would enable changes of urban areas from other land use/land cover surfaces to be detected. These periods were chosen because the negative impacts of urbanization in Accra such as flood disasters became serious from the 1990s and these impacts are a major challenge for developers and other stakeholders in the area. Due to the cloud-free imagery that were carefully selected and downloaded for the investigation, the analysist did not encounter the issue of scale line errors in any of the images.

8. “To ensure consistency, the collected data was cleaned and formatted.” How? Need justification.

RESPONSE: Data on topography, hydrology, land use and historical flood events are used to map flood risk. To ensure consistency, the collected data were cleaned and formatted using the Hexagon Erdas Imagine software version 15. The techniques used in Erdas Imagine to clean and format the satellite images includes Radiometric corrections which are crucial for removing the atmospheric effects and sensor errors that can result in inconsistent brightness levels in satellite images. These radiometric corrections such as histogram matching, contrast stretching, and gamma correction enhance image details and ensure consistency in radiometric values (Xu & Lenhardt, 2020). Moreover, Geometric corrections were employed to correct distortions and align images accurately with the Earth's surface. Further, the data was georeferenced, and resampled to correct geometric errors caused by sensor orientation, terrain elevation, or projection issues (Hagos et al., 2022). This ensures that images are accurately aligned and can be overlayed with other geographic datasets. Notwithstanding, the Noise and Haze Removal/reduction was also employed using the Erdas Imagine software to filter or remove noise and clouds while preserving image details. These filter help improves the visual quality of satellite images and reducing distractions caused by noise

9. Have the authors experimented with additional classification schemes for constructing a flood risk map? Currently, they assigned the five classes map. Have they assessed the impact of using additional classes? If so, do the spatial distribution of the different classes in the flood risk map and the derived results change significantly?

RESPONSE: The authors considered several scenarios regarding the categorization or the choice of the five classes for the flood risk map, even though it is the most appropriate and widely used classification scheme. Thus, when more than five classes were used, the result did not change significantly. It kept on giving the same output and results.

10. The authors should also clearly explain how they classified their flood risk map into five groups. Changing the limits of the zones can alter the entire result of their study, so a justification is needed.

RESPONSE: Creating a flood risk map requires an effective classification scheme that accurately represents the different levels of flood risk (Osman & Das, 2023). A good classification scheme should be easy to understand, based on reliable data sources, and allow for detailed analysis and decision-making (Farhadi & Mohammad, 2021). To accurately depict the flood risk levels in an area, flood risk maps are divided into different classes based on the likelihood and severity of flooding. The five classes that are commonly used in flood risk mapping are Very High, High, Moderate, Low, and Very Low (Farhadi & Mohammad, 2021). These classifications were based on the probability of flooding, the potential severity of damage from flooding, and the vulnerability of the area to flood hazards. This distinct classification increases accuracy, multiple perspectives, improved communication, accessibility, and flexibility in flood risk management. The classifications also encourages a standardization of communication between policymakers and the community, streamlines flood risk mitigation and adaptation strategies (Weday et al., 2023). For example, areas that are likely to experience frequent flooding events, such as those near rivers or in low-lying coastal regions, are typically classified as Very High or High risk. These areas are likely to experience severe damage during flood events and require significant resources to mitigate risk. Moderate risk areas, on the other hand, are those that are less likely to experience frequent flooding events but may still be vulnerable to severe flooding (Osman & Das, 2023). These areas may be located near smaller streams or rivers, or in areas with high soil saturation or poor drainage. While the likelihood of flooding is lower in these areas, property damage and public safety are still a concern. Low and Very Low risk areas are those that are typically located at higher elevations or further from rivers, streams, or coastlines. These areas are less vulnerable to flooding, but still be affected by flash floods or severe storms. Scholars argue that the use of five classes for flood risk map provides greater precision in identifying the level of flood risk in different areas (Nkwunonwo et al., 2020, Weday et al., 2023). This allows for more effective planning and implementation of mitigation and adaptation strategies. Moreover, the five classes offer multiple perspectives of the level of flood risk in different areas. This helps policymakers understand the severity and complexity of flooding, which is essential for developing efficient responses. Also, it provides a standardized communication tool for stakeholders. This helps them to understand and interpret the flood risk map, leading to a more coordinated and effective response to floods. Further, the five classes allow for a simplified version of the flood risk map that is easily accessible to different stakeholders. This makes it easier for them to understand and interpret the information provided as they do not need any technical knowledge. Additionally, Xu and Lenhardt (2020) noted that use of five classes offers flexibility in the classification of flood risk depending on specific variables such as location, severity, and frequency of flood events. It also allows for the inclusion of other factors that could influence flood risk such as infrastructure and natural barriers (Nkwunonwo et al., 2020)

11. Finally, the authors should enhance the discussion section by discussing: (i) the implications of their findings in the context of the current trend (and necessity) to evaluate flood hazards and secondary effects of a specific phenomenon. (ii) can the approach of the present paper have practical usage for planners, and how can their findings be communicated? In other words, the authors should emphasize the added value of their paper in the context of the topic

RESPONSE: The implications of this research findings in the context of the current trend would help evaluate flood hazards and secondary effects of a specific phenomenon. The increase in urban land use/land cover change from 95.51% in the period 1991-2002 to 129.14% in the period of 2002-2015 shows the level of urban physical development is high. This means that urban development has eaten up other land uses/land covers which if not critically monitored, even essential vegetation land cover land use could be replaced with buildings and urban development. It is the build space in the Greater Accra region. The pattern of urbanization is unplanned and very high, where physical developments in waterways are common and interfere with flow water in waterways The pattern of flood disasters increased from 1991 to 2015 with evidence showing two years having repeated flood events. This means implies a positive upward increase in flooding and urbanization in the study site. Thus, if quick action is not taken swiftly, a continues increase uncontrolled urban development can translate reduction in urban vegetation, physical development on waterways and other land covers. Urban planners, developers, policy makers and local communities need to make conscious efforts to deal with unregulated urbanization. The findings of this research will be disseminated through online platforms, presentations and discussed at academic and non-academic seminars.

Reviewer #2’s comments: 

The research focuses on the analysis of urbanization and flood risk in the Greater Accra Region of Ghana. The study aims to map and analyze the trends of urbanization, identify flood risk zones, and analyze the relationship between urbanization patterns and flood disaster events from 1990 to 2015. The research also aims to contribute to the understanding of urbanization and flood disaster risk in the study area and similar regions. It emphasizes the importance of effective urban planning, land use policies, and regulations to minimize the expansion of urban areas and reduce flood disaster risk. I enjoyed reading the paper. However, I have some comments for further improvement of the paper. I suggest major revision.

RESPONSE: Many thanks for the comments. We have revised the manuscript per your comments.

---

## [Decision Letter · Decision Letter 1]

18 Sep 2023

“URBANIZATION AND FLOOD RISK ANALYSIS USING GEOSPATIAL TECHNIQUES

PONE-D-23-15951R1

Dear Dr. Tankpa,

We’re pleased to inform you that your manuscript has been judged scientifically suitable for publication and will be formally accepted for publication once it meets all outstanding technical requirements.

Kind regards,

Mohammed Sarfaraz Gani Adnan, PhD

Academic Editor

PLOS ONE

Additional Editor Comments (optional):

Reviewers' comments:

Reviewer's Responses to Questions

**Comments to the Author**

1. If the authors have adequately addressed your comments raised in a previous round of review and you feel that this manuscript is now acceptable for publication, you may indicate that here to bypass the “Comments to the Author” section, enter your conflict of interest statement in the “Confidential to Editor” section, and submit your "Accept" recommendation.

Reviewer #1: All comments have been addressed

Reviewer #2: All comments have been addressed

2. Is the manuscript technically sound, and do the data support the conclusions?

Reviewer #1: Yes

Reviewer #2: Yes

3. Has the statistical analysis been performed appropriately and rigorously? 

Reviewer #1: N/A

Reviewer #2: Yes

4. Have the authors made all data underlying the findings in their manuscript fully available?

Reviewer #1: Yes

Reviewer #2: Yes

5. Is the manuscript presented in an intelligible fashion and written in standard English?

Reviewer #1: No

Reviewer #2: Yes

6. Review Comments to the Author

Reviewer #1: The authors addressed all the comments properly. I have no further observation. Therefore, the paper can published in its current form.

Reviewer #2: The authors addressed all of my comments. The paper now can be considered for publicatin in the journal.

7. PLOS authors have the option to publish the peer review history of their article (what does this mean?). If published, this will include your full peer review and any attached files.

Reviewer #1: **Yes: **Mahfuzur Rahman, PhD

Reviewer #2: No

---

## [Editor Report · Acceptance letter]

6 Oct 2023

PONE-D-23-15951R1 

“URBANIZATION AND FLOOD RISK ANALYSIS USING GEOSPATIAL TECHNIQUES 

Dear Dr. Tankpa:

I'm pleased to inform you that your manuscript has been deemed suitable for publication in PLOS ONE. Congratulations! Your manuscript is now with our production department. 

Kind regards, 

on behalf of

Dr. Mohammed Sarfaraz Gani Adnan 

Academic Editor

PLOS ONE